# New Polymethoxyflavones from *Hottonia palustris* Evoke DNA Biosynthesis-Inhibitory Activity in An Oral Squamous Carcinoma (SCC-25) Cell Line

**DOI:** 10.3390/molecules27144415

**Published:** 2022-07-10

**Authors:** Jakub W. Strawa, Katarzyna Jakimiuk, Łukasz Szoka, Krzysztof Brzezinski, Paweł Drozdzal, Jerzy A. Pałka, Michał Tomczyk

**Affiliations:** 1Department of Pharmacognosy, Faculty of Pharmacy with the Division of Laboratory Medicine, Medical University of Białystok, ul. Mickiewicza 2A, 15-230 Białystok, Poland; jakub.strawa@umb.edu.pl (J.W.S.); katarzyna.jakimiuk@umb.edu.pl (K.J.); 2Department of Medicinal Chemistry, Euroregional Center of Pharmacy, Faculty of Pharmacy with the Division of Laboratory Medicine, Medical University of Białystok, ul. Mickiewicza 2D, 15-222 Białystok, Poland; lukasz.szoka@umb.edu.pl (Ł.S.); pal@umb.edu.pl (J.A.P.); 3Department of Structural Biology of Prokaryotic Organisms, Institute of Bioorganic Chemistry, Polish Academy of Sciences, ul. Noskowskiego 12/14, 61-074 Poznań, Poland; kbrzezinski@ibch.poznan.pl (K.B.); pdrozdzal@ibch.poznan.pl (P.D.)

**Keywords:** *Hottonia palustris*, Primulaceae, polymethoxyflavones, SCC-25, HPLC-PDA

## Abstract

Four new compounds, 5-hydroxy-2′,6′-dimethoxyflavone (**4**), 5-hydroxy-2′,3′,6′-trimethoxyflavone (**5**), 5-dihydroxy-6-methoxyflavone (**6**), and 5,6′-dihydroxy-2′,3′-dimethoxyflavone (**7**), and three known compounds, 1,3-diphenylpropane-1,3-dione (**1**), 5-hydroxyflavone (**2**), and 5-hydroxy-2′-methoxyflavone (**3**), were isolated from the aerial parts of *Hottonia palustris*. Their chemical structures were determined through the use of spectral, spectroscopic and crystallographic methods. The quantitative analysis of the compounds (**1–7**) and the zapotin (**ZAP**) in methanol (**HP1**), petroleum (**HP6**), and two chloroform extracts (**HP7** and **HP8**) were also determined using HPLC-PDA. The biological activity of these compounds and extracts on the oral squamous carcinoma cell (SCC-25) line was investigated by considering their cytotoxic effects using the MTT assay. Subsequently, the most active compounds and extracts were assessed for their effect on DNA biosynthesis. It was found that all tested samples during 48 h treatment of SCC-25 cells induced the DNA biosynthesis-inhibitory activity: compound **1** (IC_50_, 29.10 ± 1.45 µM), compound **7** (IC_50_, 40.60 ± 1.65 µM) and extracts **ZAP** (IC_50_, 20.33 ± 1.01 µM), **HP6** (IC_50_, 14.90 ± 0.74 µg), **HP7** (IC_50_, 16.70 ± 0.83 µg), and **HP1** (IC_50_, 30.30 ± 1.15 µg). The data suggest that the novel polymethoxyflavones isolated from *Hottonia palustris* evoke potent DNA biosynthesis inhibitory activity that may be considered in further studies on experimental pharmacotherapy of oral squamous cell carcinoma.

## 1. Introduction

Cancer contributed to 10 million deaths in 2020. The International Agency for Research on Cancer indicates a nearly 50% increase in the incidence of carcinoma, with oral tumors being responsible for almost 380,000 cancer cases and 180,000 deaths per year [1]. The secondary metabolites of higher plants remain an important source of cytostatic and cytotoxic substances as well as non-toxic substances with preventive effects [2]. Vincristine, paclitaxel, and homoharringtonine have been successfully used in medicine, and nearly a hundred new substances are in the clinical trial stage [3]. Due to the multidirectional action of flavonoids, they are considered to be compounds that can play a large role in anticancer therapy [4,5,6]. Considering a certain range of selectivity, flavonoids, including flavones, have shown in vivo and in vitro activities against tongue squamous carcinoma. Among others, apigenin, baicalein, and naringenin have shown cell cycle inhibition in the G0/G1 phase. In contrast, flavane-3-ol derivatives such as epigallocatechin gallate (EGCG), epicatechin gallate (ECG), and epigallocatechin (EGC) have shown tumor cell growth inhibition and the activation of apoptosis [7].

*Hottonia palustris* L. (Primulaceae) (syn.: featherfoil, water violet, Wasserprimel) is a relatively undemanding semiaquatic plant that is widely distributed throughout the lowlands of Western Europe and northern Asia. It grows in calm, shallow waters, ponds, and ditches, mainly forming compact phytocenosis *Hottonietum palustris* [8]. Phytochemical and pharmacological data about *H. palustris* are scarce, and thus, this justifies the need for a more accurate investigation of the chemical constituents and the plant’s possible biological targets. The latest literature data mention the presence of saponins [9]. It has been used in folk medicine to treat various diseases, e.g., heart problems [10]. Water violet flowers are also used in the production of Bach’s medicinal essences [11]. Moreover, compounds **2–4** showed a moderate ability to inhibit tyrosinase activity (IC_50_ 120–127 µM) [12]. The results presented in this study represent an approach for searching for new anticancer agents. Therefore, the aim of this study was to take advantage of the potentialities of spectroscopic, spectrometric and crystallographic methods by applying them to the characterization and structural elucidation of flavones derivatives in *H*. *palustris* while paying attention to the determination of the methoxylation position in some compounds with the use of and 2D NMR data. To the best of our knowledge, four of them are new natural compounds occurring in the plant kingdom. The quantitative characterization of individual derivatives in extracts is completed with the bioassay-guided assessment of the cytotoxic effect and inhibition of DNA synthesis in tumor cells.

## 2. Results and Discussion

### 2.1. Structural Elucidation of Dibenzoylmethane and 5-Hydroxyflavone Derivatives

In previous analyses, the presence of numerous representatives of the polyphenol group in *H. palustris* herb extract was found via the LC-MS method, with the extract mainly including polymethoxyflavones [13,14]. The first attempts to isolate the compounds were made. To ensure high-quality spectroscopic and spectral analyses, all of the compounds were converted into their crystal forms. Then, a series of experiments was performed using compound analysis with ultraviolet light (UV-Vis), high-resolution mass spectrometry (HRMS), and X-ray diffraction assessments. The final determination of the molecule structure was carried out on the basis of one-dimensional (^1^H, ^13^C, DEPT) and two-dimensional (COSY, ROESY, HMBC, HSQC, HMQC) nuclear magnetic resonance (NMR). A summary of the signal shift values is presented in Table 1.

#### 2.1.1. 1,3-Diphenylpropane-1,3-Dione (Dibenzoylmethane) (**1**)

After recrystallization, compound **1** was obtained in two polymorphic forms: pale-yellow-colored crystals in a monoclinic polymorph and large yellow-colored crystals in their orthorhombic form. Based on the ion 223.07645 *m/z* [M-H]^−^ obtained using HRMS in the negative ionization mode, the molecular formula was established as C_15_H_12_O_2_. Ultraviolet analysis allowed the registration of two absorption maxima at 252 and 345 nm. The comparison of the NMR spectra with the literature data [15] as well as the results from the crystallographic analysis [16] made it possible to confirm that the obtained compound was 1,3-diphenylpropane-1,3-dione, which was previously found to occur as a minor constituent of licorice [17]; *Acca sellowiana* essential oil [18]. This compound is often present as in equilibrium, having two tautomeric forms (see, Figure 1)

#### 2.1.2. 5-Hydroxyflavone (Primuletin) (**2**)

Compound **2** was obtained as yellow needle crystals. Based on the obtained parent ion 237.05572 [M-H]^−^, the following molecular formula was generated: C_15_H_10_O_3_. The recorded UV spectrum (271, 336 nm) made it possible to search for the compound structure among the flavones group. This hypothesis was confirmed by the bathochromic shift (42 nm) caused by the addition of sodium methoxide (NaOMe), which proved that there is no free -OH group at the C-3 carbon. On the other hand, the complex with aluminum chloride (AlCl_3_), which is unstable in the presence of hydrochloric acid (HCl), confirmed the presence of the -OH group at C-5 (unstable bathochromic shift, 58 nm). The lack of a shift in the absorption maximum after the addition of sodium acetate ruled out the presence of a free -OH group at the C-7 carbon and ortho substitutions on the A and B rings after the impact of boric acid (H_3_BO_3_) [19]. The NMR spectra also provided fundamental information such as singlet *δ*_H_ 6.75, which corresponded to a proton on carbon atom C-3 and a value of *δ*_C_ 106.05, which is typical of flavone molecules. Moreover, the structure of isoflavone was excluded by C-2 chemical shifts at *δ*_C_ 164.55 [20,21]. The presence of the hydroxyl group at C-5 was certified by *δ*_H_ 12.58 (1H, s).

According to the information provided above as well as the results of the crystallographic analysis (see Appendix A), compound **2** was determined to be 5-hydroxyflavone (5-hydroxy-2-phenyl-4*H*-chromen-4-one), known as a primuletin, a natural product found in *Conchocarpus heterophyllus* [22], *Primula denticulate* [23], and *P. turkestanica* [24].

#### 2.1.3. 5-Hydroxy-2′-Methoxyflavone (**3**)

The isolation process led to compound **3** being obtained as pale-yellow needle crystals. The recorded UV spectrum (271, 336 nm) allowed the compound to be classified into flavones. An analysis of the changes in the UV spectrum using ionizing and complexing reagents resulted in similar conclusions as those drawn in compound **2** [19]. Based on the 267.053 *m/z* [M-H]^−^ ion peak, the molecular formula C_16_H_12_O_4_ was generated. Moreover, the apparent loss of 15 *m/z* [M-H-CH_3_]^−^ is indicative of the degradation of the methoxyl moiety. One-dimensional NMR spectra confirmed the methoxylation in ring B (δ_H_ 3.95, 1H, s; *δ*_C_ 55.67). To summarize the results obtained from the spectral and spectroscopic analyses as well as from the available scientific literature [25] and crystallographic data (see Appendix A), compound **3** was determined to be 5-hydroxy-2′-methoxyflavone (5-hydroxy-2-(2-methoxyphenyl)-4*H*-chromen-4-one). Compound **3** has previously been found in *Iris ensata* [25].

#### 2.1.4. 5-Hydroxy-2′,6′-Dimethoxyflavone (**4**)

Compound **4** was obtained as thin, white needle crystals. The registered predominant ion at 297.0772 *m/z* [M-H]^−^ (see Appendix A) meant that the molecular formula was determined to be C_17_H_14_O_5_, with an error in relation to the theoretical mass value of 1.35 ppm. Attempting to determine the structure by analyzing the changes in the UV spectrum allowed the compound to be classified as a flavone derivative (260, 327 nm) (see Appendix A) [19]. The analysis of the ^1^H and ^13^C NMR spectra in comparison to the previous structure revealed the lack of the proton signal at C-2′ and C-6′, with the simultaneous appearance of a characteristic signal for the methyl group (*δ*_H_ 3.81, 6H, s; *δ*_C_ 55.98) (see Appendix A and Table 1). This allowed us to determine the symmetrical conformation of the B ring and its substitution with two methoxyl groups on C-2′ and C-6′, respectively.

Investigation of the homonuclear through-bond (COSY) and through-space (ROESY) correlations (see Appendix A) confirmed the hypothesis that there were two -CH_3_ groups. Additionally, the protons attached to C-6-C-7-C-8 and C-3′-C-4′-C-5′, showed a direct relationship, and the ^1^H-^1^H interaction in that space attached to carbon atoms C-3′ and C- 5′ via the protons of the methyl moiety at the previously suggested site (C-2′ and C-6′). Finally, the connection of the information presented above with the correlations with the HMBC confirmed the dependence of the proton on the substituent via its correspondence with the correlating carbon (-CH_3_ δ_H_ 3.81 and C-2′, C-6′ δ_C_ 158.51) (see Appendix A). The arguments presented above were also supported by HMQC and the X-ray diffraction results (see and Appendix A). A graphic representation of this dependency is presented in Figure 2. The available literature indicates the presence of a compound exemplifying a constitutional substitution isomer (5-hydroxy-6,2′-dimethoxyflavone) [26]. Furthermore, the symmetry of the B ring is shown to comprise equivalent atoms with the signals δ_H_ 3.81 (6H, s) and δ_C_ 55.98 as well as the signals corresponding to the HC-6 proton (δ_H_ 6.8 (1H, dd, *J* = 8.03, 0.75 Hz)). In conclusion, compound **4,** which is a new natural product, was determined to be 5-hydroxy-2′,6′-methoxyflavone (5-hydroxy-2-(2,6-dimethoxyphenyl)-4H-chromen-4-one).

#### 2.1.5. 5-Hydroxy-2′,3′,6′-Trimethoxyflavone (**5**)

Compound **5** was crystallized as bright yellow cubes. The registered predominant ion 327.08874 [M-H]^−^ meant that the molecular formula was determined to be C_18_H_16_O_6,_ with an error mass value of 1.36 ppm. The obtained UV spectrum (257, 331 nm) suggested that the substance was a flavone (see Appendix A). Moreover, the results show a bathochromic shift of the I band comprising compounds **2**–**4**, resulting in the prospective increase in the oxygenation of the B-ring. On the other hand, the oxygenation pattern of the B-ring does not cause a shift in band II [19] in the MS spectrum ions (313, 297, 282 *m/z*), indicating the loss of the [M-H-CH_3_]^−^ fragment connected to the presence of three methoxy groups (see Appendix A). This assumption was confirmed by one-dimensional NMR spectra. Additionally, two doublets at *δ*_H_ 7.03 and *δ*_H_ 6.69 with the coupling constant *J* = 9.03 Hz indicated a connection between C-4′ and C-5′ with no protons in the vicinity. They submitted the asymmetric distribution of the substituents on the B ring (see Appendix A and Table 1). The ^1^H-^1^H correlation only revealed proton interactions at carbon atoms C-4′ and C-5′ in ring B. Ring A, beyond the OH group at C-5, showed no substitution (see Appendix A). The spatial proximity of the methyl groups was also indicated by the strong NOE interaction of the protons of these groups when they were substituted for carbon atoms C-3 ‘and C-6′ (see Appendix A). A detailed analysis of the data from the HSQC correlation allowed the protons directly belonging to the methyl group to be determined (*δ*_H_ 3.88, *δ*_C_ 61.48; *δ*_H_ 3.87, *δ*_C_ 56.53; and *δ*_H_ 3.77, *δ*_H_ 56.17 for substitutions at C-2′, C-3′, and C-6′, respectively) (see Appendix A). On the other hand, the analysis of the HMBC spectrum allowed the coupling between the protons of the methyl groups and the carbon atoms in the B ring to be determined (*δ*_H_ 3.88, *δ*_C_ 146.99; *δ*_H_ 3.87, *δ*_C_ 148.40; and *δ*_H_ 3.77, *δ*_C_ 151.56 for substitutions at C-2′, C-3′, and C-6′, respectively) (see Appendix A). To summarize, using the results and data obtained from the crystallographic analyses (see Appendix A), compound **5,** a new natural product, was determined to be 5-hydroxy-2′,3′,6′-methoxyflavone (5-hydroxy-2-(2,3,6-trimethoxyphenyl)-4*H*-chromen-4-one).

#### 2.1.6. 2′,5-Dihydroxy-6-Methoxyflavone (**6**)

Compound **6** was obtained as thin white needle crystals. The obtained predominant ion 283.06166 [M-H]^−^ meant that the molecular formula was determined to be C_16_ H_12_ O_5_, with an error mass value at 1.65 ppm (see. Appendix A). The obtained UV spectrum (259, 327 nm), as in the case of compounds **2–5**, confirmed that the isolated compound was a flavone with an -OH group at C-5 carbon (unstable bathochromic shift 58 nm, after AlCl_3_ addition) and a high degree of B-ring oxidation (band I = 327 nm) (see Appendix A) [19]. The ^1^H NMR analysis showed an additional -OH group (*δ* 10.13, 1H, s), one methyl group (*δ* 3.75, 3H, s) and no A and C ring substituents. Two triplets at *δ*_H_ 7.65 (*J* = 8.3 Hz) and *δ*_H_ 7.32 (*J* = 8.3 Hz) from C-7 and C-4′, respectively, indicated the proximity of other protons (see Appendix A and Table 1). Their direct connections were confirmed by the COSY correlation (see Appendix A). Moreover, using the HMQC correlation, all the protons that were directly bound to carbon were correlated (see Appendix A). Then, the determination of the methyl vincinal proton’s interaction with carbon atom (*δ*_H_ 3.75, *δ*_C_ 158.21) and the geminal interaction of the proton of both OH groups (*δ*_H_ 10.13, *δ*_C_ 156.53; *δ*_H_ 12.69, *δ*_C_ 159.95; C-2′ and C-5, respectively) allowed the final configuration to be determined (see Appendix A). The data presented above were also supported by the X-ray diffraction results (see Appendix A) and the ROESY correlation analysis (see Appendix A). The collected results allowed for compound **6** to be recognized as 2′, 5-dihydroxy-6-methoxyflavone (5-hydroxy-2-(2-hydroxy-6-methoxyphenyl)-4*H*-chromen-4-one), another new structure being described for the first time in plants.

#### 2.1.7. 5,6′-Dihydroxy-2′,3′-Dimethoxyflavone (**7**)

Compound **7** was also obtained in the form of thin white needle crystals during the recrystallization process. The molecular ion 313.07244 [M-H]^−^ meant that the molecular formula was determined to be C_17_H_14_O_6,_ with an error in relation to the theoretical value at 2.4 ppm (see Appendix A). The UV spectrum indicated that it was a flavone derivative (258, 333 nm). The value of band I in the spectrum indicated a high degree of B ring oxidation (see Appendix A). Additionally, in the mass spectrum, a 2-fold loss of 15 *m/z* relative to the parent ion suggests the cleavage of the 2 methoxy groups. Moreover, the presence of a 255 *m/z* ion that is typical of dihydroxyflavone formed via the complete degradation of the CH_3_ groups (see Appendix A). Further, the ^1^H spectrum showed signals from two asymmetrically arranged methyl groups (*δ* 3.77, 3H, *s; δ* 3.78, 3H, *s*), and an -OH group (*δ* 9.74, 1H, *s*) allowed locating them in the B ring as opposed to in the -OH group (*δ* 12.65, 1H, *s*) attached to C-5 (see Appendix A and Table 1). A detailed analysis of the correlation spectra provided specification of the structure configuration. COSY analysis confirmed the lack of A ring substitution. More importantly, it proved that correlation could only be observed between the protons at the HC-4′ and HC-5′ carbons (see Appendix A). On the other hand, ROESY confirmed proton interaction in the methyl attached at C-3′ and the proton at HC-4′, while HMBC analysis confirmed the location of both methyls (*δ*_H_ 3.77, *δ*_C_ 147.45*; δ*_H_ 3.78, *δ*_C_ 145.15, C-2′ and C-3′, respectively) (see Appendix A). Taking the above results, HSQC analysis (see Appendix A) and the crystallographic analysis data into consideration (see Appendix A), we concluded that compound **7** is a new structure of natural origin called 5, 6′-dihydroxy-2′,3′-dimethoxyflavone (5-hydroxy-2-(6-hydroxy-2,3-dimethoxyphenyl)-4*H*-chromen-4-one). The structures of the analyzed substances are shown in Figure 3.

### 2.2. Quantification of Flavones (1–7) in Selected Extracts

The secondary metabolites obtained from the flavonoid group were quantified using a liquid chromatograph equipped with a photodiode array detector (HPLC-PDA). The applied C8 column allowed for the reduction of the organic mobile phase modifier used during the analysis as well as the reduction of the retention time of the compounds, which made the method environmentally friendly and cost-effective. In the selected extracts (**HP1**, **HP6–8**), the content of isolated compounds (**1–7**) as well as the zapotin (**ZAP**) found in the raw material in previous analyses [11,25] was determined (see Table 2 and Appendix A). The highest content was found in **HP6** (338 µg/mg), followed by **HP7** (76.31 µg/mg). The lowest flavonoid content was determined in **HP8** (7.99 µg/mg). Such a dependence confirms the legitimacy of isolation from the **HP6** extract and proves that the isolated non-polar compounds will go to the extracts from the initial phase of extraction, remaining almost completely extracted from the plant material.

Moreover, **ZAP** (3.84–154.77 µg/mg) was dominant in all fractions, followed by compound **3** (0.86–58.82 µg/mg) in **HP1**, **HP6,** and **HP8** and compound **6** (10.21 µg/mg) in **HP7**.

### 2.3. Cell Viability

An MTT assay for each sample was performed for a preliminary estimation of the cytotoxic effects of the extracts **HP1** and **HP6**–**HP****8** (6.25–200 µg/mL), compounds **1**–**7** and **ZAP (**6.25–200 µM) in human tongue squamous carcinoma cells SCC-25 and normal fibroblasts after 24 h and 48 h of incubation (see Table 3). The ability of the tested samples to induce cytotoxic effects in SCC-25 cells was only observed for the selected compounds. It is worth mentioning that there are several reports with information on the reducing effect of plant metabolites on the conversion of tetrazolium dye to formazan. However, the reduction is connected to the presence of a catechol group as well as free OH groups at C-3 and C-5′ [27,28]. The examined compounds do not have groups that can disturb the reliability of the measurements.

The MTT study showed that SCC-25 cell viability was inversely proportional to the applied concentration of all of the tested compounds. The cytotoxic potential is expressed as a median inhibitory concentration (IC_50_) value, where the studied extracts and compounds inhibited the viability of the SCC-25 cells in a dose-dependent manner. The highest activity was observed after treatment with compounds **1** and **7** as well as with **ZAP** after both 24 h and 48 h of incubation (see Table 3). The IC_50_ values of the other compounds were >200 µM (highest tested concentration). The IC_50_ values after 48 h of incubation for some extracts were worth emphasizing (HP6, IC_50_ = 14.9 µg/mL; HP7, IC_50_ = 16.7 µg/mL) as they reached half of the effectiveness cisplatin, which was used as a positive control (**cPT**, IC_50_ = 7.58 µg/mL). It is worth noting that the activity of all of the tested samples was higher after 48 h of incubation than after 24 h. Moreover, it can be assumed that the cytotoxic effect of the extracts correlates with the content of all of the flavones present (**HP6** > **HP7** > **HP1** > **HP8**) as well as the **ZAP** and compound **7** content. The viability of the fibroblast under the influence of **HP1** and **HP6** and **1**, **7**, and **ZAP** was significantly higher than it was in the SCC-25 cells (above 200), and for **HP7** it was 58.9 ± 2.94 µg/mL.

The morphological profile of SCC-25 squamous carcinoma cells after 24 h and 48 h of incubation with **HP1**, **HP6** and **HP7** at the concentration of 100 µg/mL, and **1**, **7**, **ZAP**, and **cPT** are shown in Figure 4 and Figure 5. We observed slope in cell adhesion and decreased the number of cells in the treatment groups compared to the untreated group, displaying the inhibition of cell proliferation.

In order to determine the potential mechanism of action, the level of the pro-apoptotic proteins was determined via the Western blot method. The obtained results did not indicate the presence of this kind of activity (unpublished data). It is confirmed by microphotographs (Figure 4 and Figure 5) that show no typical morphological features of apoptosis, in particular cell membrane blebbing and formation of apoptotic bodies. Considering this fact, an evaluation determining whether the cytotoxicity of the samples was the result of their influence on the biosynthesis of cell DNA was carried out by assessing the capability of intracellular thymidine incorporation. A downregulation in the DNA biosynthesis in the SCC-25 cells was observed for all the tested samples (see Figure 6 and Figure 7).

Relating these effects to the isolated compounds, dose-dependency appears at concentrations above 25 µM. Most importantly, the potency of **ZAP** after 48 h shows an effect similar to that of the positive control, **cPT**. The assessed cytotoxic effect and the influence on DNA synthesis may suggest that the mechanism of action in the samples is based on the influence on cell division via an arrest of the division in the S phase of the cell cycle. Our results revealed that the viability of cancer cells is the most strongly affected by **ZAP** compared to any of the other flavonoids isolated from *H*. *palustris*. Zapotin (**ZAP**), which has been tested on numerous biological models, including on the human bladder carcinoma cell (T24), human liver cancer cell (HepG2), and human leukemia cell (HL-60) lines, showed suppression in the G2/M phase of the cycle [29]. In addition, the current reports on the polymethoxyflavones confirm the cystostic mechanism of action consisting in arrest of the cell cycle [30]. Although the effect on different lines cannot be directly correlated, compound **1**, identified as dibenzoylmethane, has already shown cell-cycle deregulation in melanoma cells [31]. In light of the available literature and the results obtained, flavonoids show the strongest multidirectional action for use as anticancer agents [32].

To the best of our knowledge, the results presented above are the first detailed report comparing nonpolar extracts or compounds from *H. palustris* and describing their activity on the oral squamous carcinoma cell line (SCC-25). The obtained structures bring new light to the derivatives of secondary metabolites from the polymethoxyflavones group and extend to possible directions for compound biosynthesis in plant tissues. Moreover, it seems correct to suppose that the chemotaxonomic markers present in the Primulaceae family species, such as primuletin [21,22], can be extended with another potential chemophenetic factor such as zapotin [31].

## 3. Materials and Methods

### 3.1. Chemicals and General Methods

Petroleum, *n*-hexane, chloroform, and methanol were purchased from Avantor (Gliwice, Poland). Sephadex LH_20_ was purchased from GE Healthcare (Uppsala, Sweden). The adsorbent used for LPLC, silica gel 60 (0.063–0.2 mm), and a TLC aluminum plate coated with silica gel 60 F₂₅₄ were purchased from Merck (Darmstadt, Germany). Formic acid was purchased from Sigma Aldrich Co. (St. Louis, MO, USA). Tetrahydrofuran HPLC grade was purchased from Honeywell (Seelze, Germany). Acetonitrile Optima (LC/MS grade) was purchased from Fisher Scientific (Loughborough, UK). Ultrapure water was obtained using the POLWATER DL3-100 system (Kraków, Poland). The cell viability and proliferation assays were performed on the human tongue squamous cell line (SCC-25). The cells were obtained from ATCC (CRL-1628, American Type Culture Collection, Manassas, VA, USA) and cultured in a 1:1 mixture of DMEM (Dulbecco’s minimal essential medium) and Ham’s F12 medium (Thermo Fisher Scientific, Waltham, MA, USA, 31330038) supplemented with 400 ng/mL hydrocortisone (Sigma-Aldrich, Burlington, MA, USA, H0135), 10% FBS (fetal bovine serum, Thermo Fisher Scientific, Waltham, MA, USA, 10270106), and penicillin–streptomycin solution (Thermo Fisher Scientific, Waltham, MA, USA, 15140122). Furthermore, for the cytotoxicity assay, human skin fibroblasts (CCD25Sk) cultured in DMEM (Thermo Fisher Scientific, Waltham, MA, USA, 41966029) supplemented with 10% FBS and penicillin-streptomycin solution were used. The cells were incubated at 37 °C in the air with 5% CO_2_. PBS (14190136) was purchased from Thermo Fisher Scientific (Waltham, MA, USA,). MTT (M5655), SDS (L3771), and cisplatin (cPT), which was used as a positive control (P4394), were purchased from Sigma-Aldrich (Burlington, MA, USA). Zapotin was purchased from Biosynth-Carbosynth (Compton, UK). The [^3^H]-thymidine (#MT6037) for the proliferation assay was obtained from Hartmann Analytic (Braunschweig, Germany). Extract preparation was carried out using an ultrasonic bath (40 kHz, Sonic-5, Polsonic, Warsaw, Poland). Solvent residues were removed by distillation (Rotavapor R-215 coupled with vacuum controller V-855 (Büchi, Flawil, Switzerland) and freeze-drying (Lymph-lock, Labconco, Kansas City, MO, USA). The HPLC–PDA analyses were performed using an Agilent Infinity apparatus equipped with a 1260 high pressure binary pump, a 1260 ALS sampler, a 1260 TCC column oven, and a 1290 photodiode array detector (Agilent Technologies, Santa Clara, CA, USA). UV spectra were measured with an Analytic Jena SPECORD 200 Plus instrument (Jena, Germany). Melting points were obtained using a BÜCHI 535. NMR spectra were recorded on a Thermo Fisher Scientific Bruker Avance II 400 spectrometer (Billerica, MA, USA). High-resolution mass spectra were recorded using a 6230 TOF mass spectrometer (Agilent Technologies, Santa Clara, CA, USA). Specific rotation [α] was recorded using a P-2000 digital polarimeter (Jasco, Hachioji, Tokyo, Japan) at 25 ± 0.5 °C and at the sodium D line. 

### 3.2. Plant Material

*Hottonia palustris* herb (HP) was collected at the turn of April and May 2019 before flowering from a wild habitat in the wetlands of the Puszcza Knyszyńska (53°17′13.2″ N, 22°53′42.0″ E). Taxonomic identification was carried out based on the scientific botanical literature [33]. A voucher specimen (No. HP-17040) was deposited in the Herbarium of the Department of Pharmacognosy Medical University of Białystok, Poland.

### 3.3. Preparation of the Extracts

Immediately after harvesting, the plant material was dried under shade and air ventilation at ambient temperature (1100 g). The ultrasound-assisted extraction technique was used to obtain the crude methanol extract **HP1**. Milled and accurately weighed plant material (15 g) was extracted five times at 40 ± 2 °C with 0.05 L of methanol. Subsequently, the extract was centrifuged, filtered, concentrated under the vacuum at 40 ± 2 °C, and lyophilized. The continuous extraction technique in the Soxhlet apparatus was used to prepare the lipophilic extracts. Milled and accurately weighed HP herb (1160 g) was exhaustively extracted with petroleum. The solvent was removed under a vacuum at a controlled temperature (40 ± 2 °C) to produce a crude extract (**HP6**, 1.5 L × 45, yield 75 g). After the solvent was evaporated, the plant material was etched with chloroform. Then, the solvent was removed under a vacuum at a controlled temperature (40 ± 2 °C) to produce a crude extract (**HP7**, 1.5 L × 30, yield 28 g). Then, the purified plant material was subjected to exhaustive extraction with methanol under reflux. Afterwards, the lipid compounds were discarded, and the plant sample was subjected to exhaustive extraction with methanol and then with 50% methanol solution (4.5 L × 20, each). Subsequently, the combined methanol extracts were evaporated from the solvent at a lowered pressure, dissolved in water, and partitioned into the chloroform phase (**HP8**, 0.1 L × 45). The water residue was lyophilized and intended for further research. Each time, the efficiency of the extraction process was controlled using thin-layer chromatography. All of the extracts were suspended in water, frozen and lyophilized.

### 3.4. Isolation Procedures

The low-pressure liquid chromatography (LPLC) technique was used as a preliminary isolation step. After conditioning, the elution was carried out on a Sephadex LH_20_ column in an isocratic with a mixture of chloroform and methanol (3:2, *v*/*v*). The 18 fractions were collected. After verification via the TLC technique (silica gel, solvents *n*-hexane: ethyl acetate (8:2, *v*/*v*)) and combination, five fractions were finally obtained. Then, a fourth, flavonoid-rich fraction was separated on an LPLC column packed with silica gel. Elution was performed with an *n*-hexane and ethyl acetate mixture with gradually increasing polarity. The 100 fractions were collected. After TLC analysis (silica gel plate, solvents *n*-hexane: ethyl acetate (8:2, *v*/*v*)), similar fractions were pooled. Finally, 21 fractions were obtained. As the solvent was allowed to evaporate under low pressure at room temperature, the compounds crystallized. Thus, compound **1** (425 mg), thereupon compound **2** (89 mg), was eluted one by one with *n*-hexane: ethyl acetate at a ratio of 95:5 (*v*/*v*). Then, compound **1** required recrystallization in pure hexane [16]. Compound **3** (729 mg) and compound **4** (480 mg) crystallized from the fraction and were eluted one by one at a ratio of 8:2 (*v*/*v*). Compounds **6** (130 mg) and **7** (white thin needle, 170 mg) were eluted one by one at a ratio of 7:3 (*v*/*v*). The dry residue of the fractions was recrystallized in a mixture of dichloromethane: ethanol (9:1, *v*/*v*). Thereafter, a residue was obtained from the fraction collected at a ratio of 6:4 (*v*/*v*), which was dissolved with a mixture of hot ethanol: water (98:2, *v*/*v*). As the solution gradually cooled, compound **5** (110 mg) crystallized.

### 3.5. Crystallographic Analysis

Crystals of compounds **2–7** were mounted on a loop with paraffine oil. The X-ray diffraction data were measured using Cu Kα radiation at 100 K on a SuperNova diffractometer (Rigaku, Akishima-shi, Tokyo, Japan) equipped with either a CCD detector (**2**–**4**) or XtalLAB Synergy (Rigaku, Akishima-shi, Tokyo, Japan) with a Hybrid Pixel two-dimensional detector HyPix-6000HE (**5**–**7**). All of the data were integrated and scaled using the CrysAlisPro software package (Rigaku, Akishima-shi, Tokyo, Japan). The crystal structures were determined using direct methods according to the OLEX2 [34] graphical interface with SHELXT [35] and were refined with SHELXL [36]. All of the non-hydrogen atoms were refined anisotropically using the full-matrix least-squares method. All of the hydrogen atoms were initially located on electron-density difference maps. The aromatic hydrogen atoms were constrained to idealized positions with C-H = 0.95 Å and U_iso_(H) = 1.2U_eq_(C), whereas the methyl hydrogen atoms were constrained with C-H = 0.98 Å and U_iso_(H) = 1.5U_eq_(C). The hydroxyl hydrogen atoms were freely refined. PLATON software [37] was used to validate the final crystallographic data. The CCDC entries: 2143620 (**2**), 2143614 (**3**), 2143617 (**4**), 2143615 (**5**), 2143619 (**6**), and 216092 (**7**), and contain the supplementary crystallographic data for this publication. These data can be obtained free of charge from The Cambridge Crystallographic Data Center, see [38]. The final data sets and refinement statistics for all of the crystal structures are reported in the Appendix A.

### 3.6. Compound Identification

1,3-diphenylpropane-1,3-dione (**1**): pale-yellow crystals, UV λ_max_ nm: 252, 345; +NaOMe: 240, 351; +AlCl_3_: 267, 364; +NaOAc: 251, 344; +H_3_BO_3_: 252, 343. HRMS *m/z* = 223.07645 [M-H]^−^ (calculated for C_15_H_12_O_2_), difference = 3.2 ppm. Purity 98% by HPLC. For X-ray diffraction data, see [16]. For NMR spectral data, see [15].

5-hydroxyflavone (**2**): yellow needle, UV λ_max_ nm: 271, 336; +NaOMe: 276, 383; +AlCl_3_: 293, 397; +NaOAc: 270, 336; +H_3_BO_3_: 271, 336. HRMS *m/z* = 237.05572 [M-H]^−^ (calculated for C_15_H_10_O_3_), difference = 5.52 ppm. Purity 98% by HPLC. X-ray diffraction data have been provided in Appendix A. For NMR spectral data, see [21].

5-hydroxy-2′-methoxyflavone (**3**): pale-yellow needle crystals, UV λ_max_ nm: 260, 327; +NaOMe: 264, 371; +AlCl_3_: 267, 388; +NaOAc: 261, 327; +H_3_BO_3_: 260, 336. HRMS *m*/*z* = 267.06753 [M-H]^−^ (calculated for C_16_H_12_O_4_), difference = 2.2 ppm. Purity 98% by HPLC. X-ray diffraction data have been provided in Appendix A. For NMR spectral data, see [25].

5-hydroxy-2′,6′-dimethoxyflavone (**4**): white thin needle, (mp.: 173 °C); [α]_D_: -1.47 (DMSO; c = 0.1); UV λ_max_ nm: 260, 327; +NaOMe: 264, 371; +AlCl_3_: 267, 388; +NaOAc: 261, 327; +H_3_BO_3_: 260, 336 (see Appendix A). HRMS *m/z* = 297.0772 [M-H]^−^ (calculated for C_17_H_14_O_5_), difference = 1.35 ppm (see Appendix A). Purity 98% by HPLC. X-Ray diffraction data have been provided in Appendix A. For NMR spectral data, see Table 1 and Appendix A.

5-hydroxy-2′,3′,6′-trimethoxyflavone (**5**): bright yellow cubes (mp.: 173 °C); [α]_D_: 0.56 (DMSO; c = 0.1); UV λ_max_ nm: 257, 331; +NaOMe: 267, 372; +AlCl_3_: 268, 387; +NaOAc: 258, 331; +H_3_BO_3_: 257, 331 (see Appendix A). HRMS *m/z* = 327.08874 [M-H]^−^ (calculated for C_18_H_16_O_6_), difference = 1.36 ppm (see Appendix A). Purity 98% by HPLC. X-ray diffraction data have been provided in Appendix A. For NMR spectral data, see Table 1 and Appendix A.

2′, 5-dihydroxy-6-methoxyflavone (**6**): white thin needle, (mp.: 238 °C); [α]_D_: -0.21 (DMSO; c = 0.1); UV λ_max_ nm: 259, 327; +NaOMe: 268, 365; +AlCl_3_: 267, 385; +NaOAc: 258, 327; +H_3_BO_3_: 259, 364 (see Appendix A); HRMS *m/z* = 283.06166 [M-H]^−^ (calculated for C_16_H_12_O_5_), difference = 1.65 ppm (see Appendix A). Purity 98% by HPLC. X-ray diffraction data have been provided in Appendix A. For NMR spectral data, see Table 1 and Appendix A.

5, 6′-dihydroxy-2′,3′-dimethoxyflavone (**7**): white thin needle, (mp.: 200 °C); [α]_D_: -0.21 (DMSO; c = 0.1); UV λ_max_ nm: 258, 333; +NaOMe: 243, 369; +AlCl_3_: 268, 387; +NaOAc: 258, 332; +H_3_BO_3_: 257, 333 (see Appendix A); HRMS *m/z* = 313.07244 [M-H]^−^ (calculated for C_17_H_14_O_6_), difference = 2.4 ppm (see Appendix A). Purity 98% by HPLC. X-Ray diffraction data have been provided in Appendix A. For NMR spectral data, see Table 1 and Appendix A.

### 3.7. Method Development and Flavonoid Quantification in H. palustris by HPLC-PDA

The chromatographic method developed here was validated according to the guidelines outlined by the International Conference on Harmonisation (ICH) [39]. The parameters are summarized in Appendix A.

#### 3.7.1. Standard Preparation

Accurately weighed 1 mg of compounds **1**–**7** and ZAP was dissolved in 1 mL DMSO and filtered through a 0.2 µm PVDF syringe filter into HPLC vials. Then, standard solutions were used to prepare the stock solution. In the next step, five-step mixtures were prepared using the multiple dilution method for the calibration curve. Measurements were started immediately after the solutions were prepared.

#### 3.7.2. Sample Preparation

Accurately weighted extract samples were dissolved in DMSO. Then, they were diluted with the initial mobile phase to 10 mg/mL (*m*/*v*) (**HP1**), 1 mg/mL (*m*/*v*) (**HP6**), 5 mg/mL (*m*/*v*) (**HP7**), and 20 mg/mL (*m*/*v*) (**HP8**) and filtered through a 0.2 µm PVDF syringe filter into HPLC vials.

#### 3.7.3. Analysis Conditions

The HPLC analyses were carried out on a reverse phase Kinetex XB-C8 column (batch No.: 5606-0137, 150 × 2.1 mm, 1.7 µm; Phenomenex, Torrance, CA, USA) and were protected by the precolumn. Mobile phase A was a mixture of 2% formic acid in water and tetrahydrofuran (8:2, *v*/*v*), and mobile phase B was a mixture of 2% formic acid in acetonitrile and tetrahydrofuran (8:2, *v*/*v*). A three-step gradient solvent system, 0–3 min, 20% B; 3–30 min, 20–37% B; and 30–35 min, 37% B, was used with 10 min equilibration. The flow rate was 150 µL/min. The column was thermostated at 25 ± 0.5 °C. The volume of the injected samples was 1 µL. UV-Vis spectra were recorded over a range of 190–600 nm, and chromatograms were acquired at 258 nm (**ZAP**, compounds **4**–**7**), 270 nm (compounds **2**, **3**), and 345 nm (compound **1**) wavelengths, which matched the value of the most intense absorption peak. For every tested compound, the calibration curve was obtained by plotting the peak areas versus the number of standards. The contents of the compounds in the samples were calculated using the regression parameters of the calibration curves.

#### 3.7.4. Specificity, Linearity and Range

Specificity was tested by comparing the retention times, peaks purity, and UV spectra of the compounds in the extracts with standards. The analysis of the obtained UV chromatograms made it possible to exclude the possibility of co-elution (see Appendix A). To evidence a linear relationship between the compound signals and the compound concentration, determinations were performed for five concentrations in triplicate. The working range was defined by the restrictive limit of quantification and the highest concentration allowing for linearity.

#### 3.7.5. Accuracy

Accuracy was assessed by testing the recovery. The ratio of the known added concentration of standard compound to the concentration of standard compound calculated based on the calibration curve developed with the HPLC-PDA method was determined and expressed as a percentage of recovery.

#### 3.7.6. Precision

Precision was assessed by testing the repeatability of six independent sample solutions (intraday) and by carrying out intermediate precision analysis on three independent sample solutions on different days (interday).

#### 3.7.7. Limit of Detection (LOD) and Limit of Quantification (LOQ)

According to the guidelines outlined by the ICH [39], we have adopted the standard deviation value of the intercept (*σ*) divided by the slope (*s*) of the calibration curves multiplied by 3.3 and have then multiplied for limit of detection (LOD) and multiplied by 10 to determine the limit of quantification; see Equation 1.
(1)$$\mathrm{LOD}=\frac{3.3\sigma }{s}\;\mathrm{LOQ}=\frac{10\sigma }{s}$$

### 3.8. Biological Assays

#### 3.8.1. Cell Viability Assay

A preliminary assessment of the cytotoxicity of **HP1** and **HP6**-**HP8**, all of the isolated compounds **1**–**7**, and the found in **ZAP** from *H. palustris* on SCC-25 and normal fibroblast (PCS-201-012) cells using MTT (methyl thiazolyl tetrazolinum) was conducted according to Carmichael’s method with some modifications [40]. The cell viability assay leans on the transformation of yellow tetrazolinum bromide MTT solution to the purple derivative of formazan via succinate dehydrogenase, an enzyme present in viable cells. Cells were cultured (1 × 10^4^ per well) on a 96-well plate to receive 70% of their confluency. After 24 h and 48 h of incubation with the studied samples (prepared as DMSO solutions directly before the assay), the cells were washed with PBS (phosphate-buffered saline). Subsequently, the MTT solution was added, and the plates were incubated at 37 °C for 2 h. Then, formazan crystals were dissolved in 200 μL of DMSO with Sorensen’s glycine buffer and were shacked for 10 min. The absorbance was measured at a 570 nm wavelength using a microplate reader (EPOCH 2, BioTek, Winooski, VT, USA). The extracts and compounds were used at a final DMSO concentration of no more than 0.1% (*v*/*v*) in each well. The **HP1** and **HP6**-**HP8** extracts were analyzed at the following concentrations: 6.25–200 µg/mL, while compounds **1**–**7**, **ZAP** and **cPT** which was used as a positive control, were analyzed at 6.25–200 µM.

#### 3.8.2. Proliferation Assay

The proliferation of the tested cells was estimated as [^3^H]-thymidine incorporation into the DNA of the cells treated with the studied samples [40]. Firstly, cells (1 × 10^4^ per well) were placed in 24-well plates and were allowed to adhere for 24 h. Afterwards, the cells were treated with 1 mL of growth medium containing 0.5 μCi [^3^H]-thymidine and various concentrations of each sample (**HP1** and **HP6**-HP7; compounds **1** and **7**; and **ZAP**). **cPT** was used as a positive control. After incubation for 24 h and 48 h at 37 °C, the cells were rinsed with PBS and solubilized with 1 mL of 0.1 M NaOH (sodium hydroxide) containing 1% SDS (sodium dodecyl sulfate). Then, radioactivity was recorded in a Liquid Scintillation Analyzer (Perkin-Elmer, Waltham, MA, USA) after correction to 9 mL with scintillation liquid.

### 3.9. Statistical Analysis and Software

Mass Hunter Qualitative Analysis 10.0 (Agilent, Santa Clara, CA, USA) analysis was used to complete and process the chromatographic data. MS Excel 2016 with the Data Analysis add-on was used for statistical analysis and for regression calculation (Microsoft, Redmond, WA, USA). Linear regression parameters for the standard curve were determined using ANOVA. The IC_50_ and standard deviation values were calculated using GraphPad Prism 9 software (GraphPad Software, San Diego, CA, USA).

## Figures and Tables

**Figure 1 molecules-27-04415-f001:**
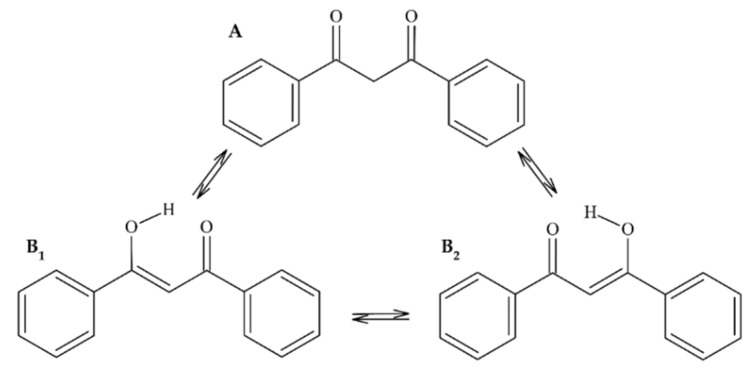
Chemical structure of compound **1.** The structures represent the tautomeric keto (**A**) and enol (**B_1_**, **B_2_**) forms in equilibrium.

**Figure 2 molecules-27-04415-f002:**
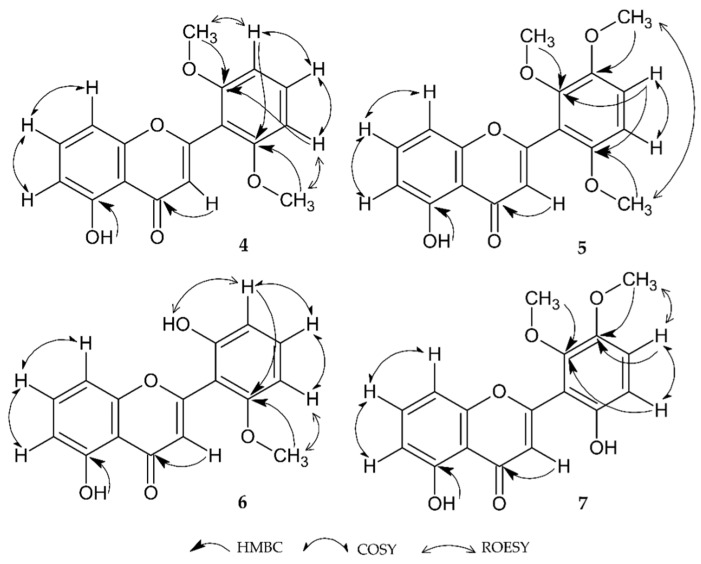
Essential correlation for the structural elucidation of compounds **4**–**7**.

**Figure 3 molecules-27-04415-f003:**
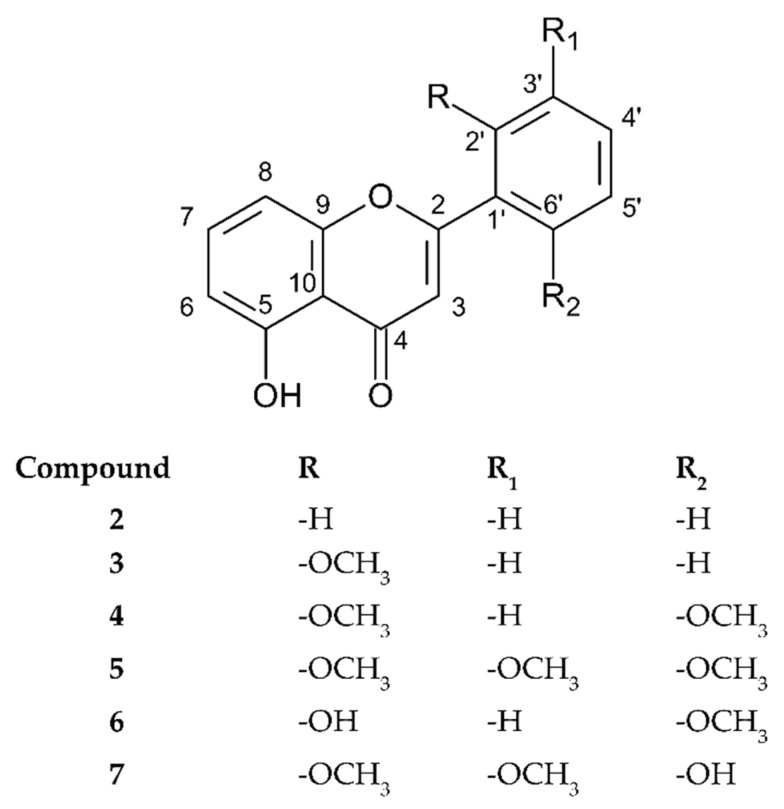
Chemical structures of compounds **2–7**.

**Figure 4 molecules-27-04415-f004:**
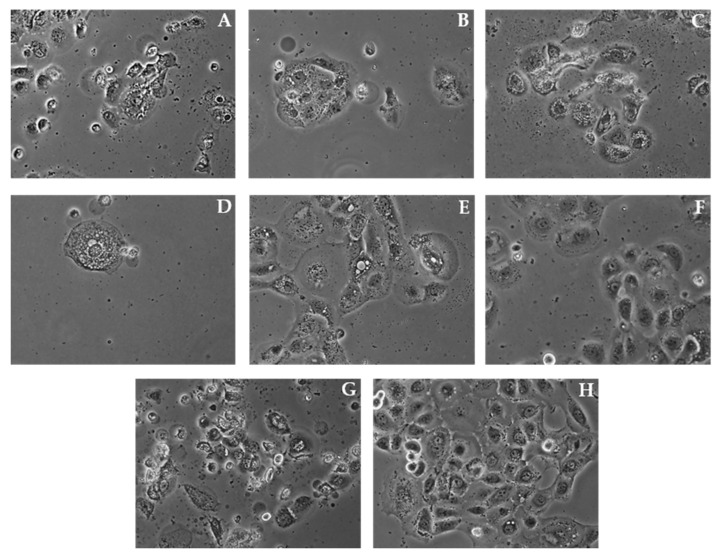
Morphological profile of SCC-25 cells after 24 h of incubation with **HP1** (**A**), **HP6** (**B**), and **HP7** (**C**) at the concentration of 100 µg/mL and with **1** (**D**), **7** (**E**), **ZAP** (**F**), and **cPT** (**G**) at 100 µM concentrations compared the untreated control cells (**H**).

**Figure 5 molecules-27-04415-f005:**
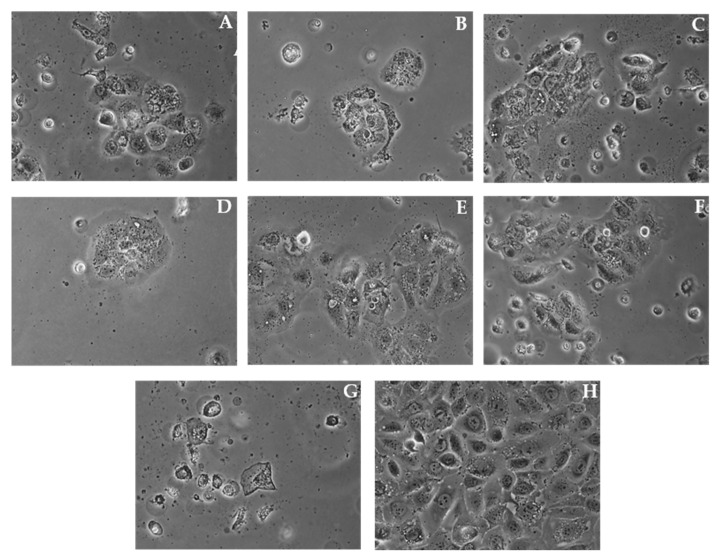
Morphological profile of SCC-25 cells after 48 h of incubation with **HP1** (**A**), **HP6** (**B**), and **HP7** (**C**) at the concentration of 100 µg/mL and with **1** (**D**), **7** (**E**), **ZAP** (**F**), and **cPT** (**G**) at 100 µM concentrations compared to the untreated control cells (**H**).

**Figure 6 molecules-27-04415-f006:**
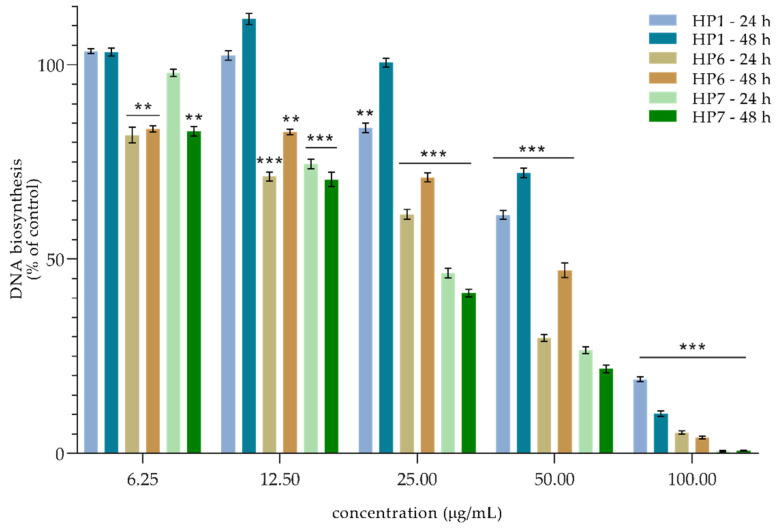
The decreasing effects of **HP1**, **HP6**, and **HP7** extracts (µg/mL) on DNA biosynthesis in SCC-25 cells after 24 h and 48 h of incubation. Mean percentage from at least three independent experiments carried out in duplicate are presented. ** *p* < 0.01 versus control group, *** *p* < 0.001 versus control group.

**Figure 7 molecules-27-04415-f007:**
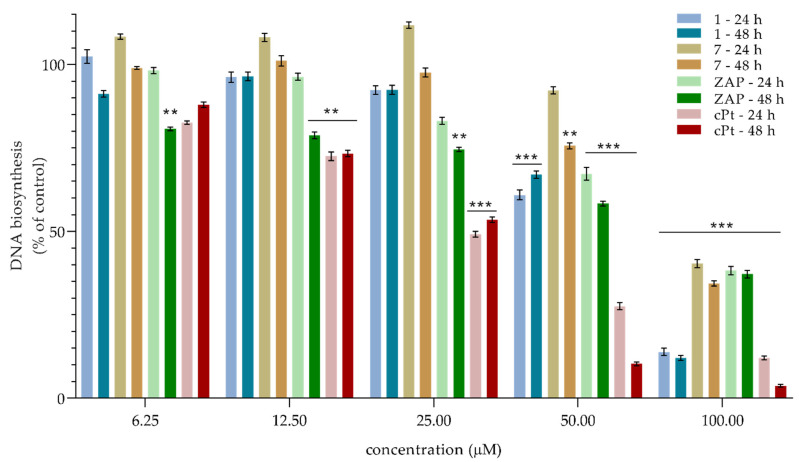
The decreasing effects of **1**, **7**, zapotin (**ZAP**)**,** and **cPT** (µM) on DNA biosynthesis in SCC-25 cells for 24 h and 48 h of incubation. Mean percentages from at least three independent experiments conducted in duplicate are presented. ** *p* < 0.01 versus control group, *** *p* < 0.001 versus control group.

**Table 1 molecules-27-04415-t001:** ^1^H and ^13^C spectral data of compounds **4–7** (400 Hz for ^1^H and 100 Hz for ^13^C spectrum, *δ* in ppm, *J* in Hz).

C No.	4 ^A^		5 ^A^		6 ^B^		7 ^B^	
	*δ* C	*δ* H	*δ* C	*δ* H	*δ* C	*δ* H	*δ* C	*δ* H
**2**	162.15	-	161.71	-	163.08	-	162.61	-
**3**	113.47	6.36, *s*	113.13	6.38, s	112.68	6.4, s	112.34	6.46, s
**4**	183.74	-	183.55	-	183.01	-	182.91	-
**5**	160.83	-	160.86	-	159.96	-	159.97	-
**6**	110.76	6.8 (dd, *J* = 8.03, 0.75)	110.95	6.81 (dd, *J* = 8.28, 0.75)	110.76	6.83 (dd, *J* = 8.5, 0.85)	110.86	6.84 (d, *J* = 8.28)
**7**	134.9	7.51 (t, *J* = 8.28)	135.07	7.52 (t, *J* = 8.28)	135.91	7.65 (t, *J* = 8.3)	136.0	7.67 (t, *J* = 8.5)
**8**	107.26	6.91 (dd, *J* = 8.50, 1.0)	107.15	6.91 (dd, *J* = 8.53, 1.0)	107.46	7.05 (dd, *J* = 8.5, 0.8)	107.44	7.08 (d, *J* = 8.28)
**9**	157.53	-	157.29	-	156.82	-	156.64	-
**10**	110.76	-	111.01	-	110.05	-	110.04	-
**5-OH**	-	12.69, s	-	12.66, s	-	12.66, s	-	12.65, s
**1′**	111.04	-	117.18	-	109.01	-	115.02	-
**2′**	158.5	-	146.99	-	156.53	-	147.45	-
**3′**	103.93	6.65 (d, *J* = 8.53)	148.40	-	108.74	6.62 (dd, *J* = 8.3, 1.8)	145.15	-
**4′**	132.41	7.42 (t, *J* = 8.5)	115.49	7.03 (d, *J* = 9.03)	132.49	7.32 (t, *J* = 8.3)	117.22	7.12, (d, *J* = 9.03)
**5′**	103.93	6.65 (d, *J* = 8.53)	106.16	6.69 (d, *J* = 9.03)	102.23	6.62 (dd, *J* = 8.3, 1.8)	110.83	6.7, (d, *J* = 9.03)
**6′**	158.5	-	151.56	-	158.21	-	149.47	-
**2′-OH**	-	-	-	-	-	10.13, s	-	-
**6′-OH**	-	-	-	-	-	-	-	9.74, s
**2′-OCH_3_**	55.98	3.81, s	61.48	3.88, s	-	-	60.94	3.77, s
**3′-OCH_3_**	-	-	56.53	3.87, s	-	-	56.55	3.78, s
**6′-OCH_3_**	55.98	3.81, s	56.17	3.77, s	55.95	3,75, s	-	-

A, B: the solvent used was CDCl_3_ or DMS-*d*_6_, respectively.

**Table 2 molecules-27-04415-t002:** Quantification results of the flavonoids in the extracts **HP1** and **HP6–8**.

	Content of Compounds Expressed as µg/mg of Dry Extract			
Compounds	HP1^A^	HP6^A^	HP7^A^	HP8^A^
**1**	0.81 ± 0.01	2.7 ± 0.01	4.98 ± 0.01	0.26 ± 0.01
**2**	0.29 ± 0.01	10.04 ± 0.02	1.76 ± 0.01	0.22 ± 0.01
**3**	1.31 ± 0.01	58.82 ± 0.44	6.02 ± 0.03	0.86 ± 0.01
**4**	1.05 ± 0.01	39.19 ± 0.09	8.44 ± 0.05	0.75 ± 0.01
**5**	0.67 ± 0.01	29.57 ± 0.02	5.34 ± 0.02	0.51 ± 0.01
**6**	0.9 ± 0.01	24.84 ± 0.15	10.21 ± 0.03	0.9 ± 0.01
**7**	1.26 ± 0.01	18.76 ± 0.36	7.83 ± 0.07	0.66 ± 0.01
**ZAP**	8.13 ± 0.06	154.77 ± 0.19	31.73 ± 0.13	3.84 ± 0.04
**Sum**	14.42 ± 0.01	338.68 ± 0.16	76.31 ± 0.04	7.99 ± 0.01

A: the content expressed as mean with standard deviation.

**Table 3 molecules-27-04415-t003:** IC_50_ of viability of SCC-25 human tongue squamous and fibroblast cells treated for 24 h and 48 h with different concentrations of the extracts **HP1** and **HP6**-**HP8** and compounds **1–7** isolated from *H. palustris* and zapotin (**ZAP**).

	24 h		48 h	
Sample	SCC-25	Fibroblasts	SCC-25	Fibroblasts
**HP1 ^A^**	37.3 ± 1.86	>200	30.3 ± 1.15	>200
**HP6 ^A^**	50.88 ± 2.54	>200	14.9 ± 0.74	>200
**HP7 ^A^**	48.4 ± 2.40	>200	16.7 ± 0.83	58.90 ± 2.94
**HP8 ^A^**	>200	-	>200	-
**1 ^B^**	39.98 ± 1.89	>200	29.1 ± 1.45	>200
**2 ^B^**	>200	-	>200	-
**3 ^B^**	>200	-	>200	-
**4 ^B^**	>200	-	>200	-
**5 ^B^**	>200	-	>200	-
**6 ^B^**	>200	-	>200	-
**7 ^B^**	78.2 ± 3.25	>200	40.6 ± 1.65	>200
**ZAP ^B^**	64.94 ± 3.24	>200	20.33 ± 1.01	>200
**cPT ^B,C^**	26.1 ± 1.03	>200	7.58 ± 0.37	40.50 ± 2.01

A: expressed as µg/mL; B: expressed as µM; all values are represented as the mean ± standard deviation (SD) from at least three independent repeats; C: **cPT**-cisplatin as a positive control.

## Data Availability

Data are contained within the article and in the Appendix A. Further information is available upon request from the corresponding author.

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
