# Peer review of "New Polymethoxyflavones from Hottonia palustris Evoke DNA Biosynthesis-Inhibitory Activity in An Oral Squamous Carcinoma (SCC-25) Cell Line"

_molecules, 2022, doi:10.3390/molecules27144415_

Round 1

Reviewer 1 Report

Dear Authors, 

Your work is interesting and demonstrates the isolation of an important class of flavonoids. I reviewed the structures with the provided NMR and mass spectral data and I found them very well assigned. 

The whole manuscript needs minor checks for the typos.

You also should compare your MTT assay results (IC50, line 298-311) with those reported for methoxylated flavonoids published in https://doi.org/10.3390/molecules26195827 

Regards  

Author Response

A point-by-point response to the Reviewer #1

Your work is interesting and demonstrates the isolation of an important class of flavonoids. I reviewed the structures with the provided NMR and mass spectral data and I found them very well assigned.

The authors would like to thank the reviewer for favorable referee’s report.

  1. The whole manuscript needs minor checks for the typos.

RESPONSE: As Reviewer suggested, the text was carefully checked and these typos have been corrected.

  1. You also should compare your MTT assay results (IC50, line 298-311) with those reported for methoxylated flavonoids published in

https://doi.org/10.3390/molecules26195827

RESPONSE: The proposed publication discusses the effect of polymethoxyflavones on human breast, ovarian and colon adenocarcinoma. In this case, the cell cycle inhibition mechanism common to the tested methoxylated flavone derivatives was confirmed, which corresponds to the cytostatic potential. The publication was included in the manuscript (page 5, line 370) and cited as 30 (lines 358-360).

Reviewer 2 Report

In the presented MS authors investigated the effect of new polymethoxyflavones isolated from Hottonia palustris on the oral squamous carcinoma (SCC-25) cell line.

The study is developed in a current scientific field with great potential for future work.

I have some questions pointed to the authors:

1. The authors analyzed seven compounds four of them are new and three are known. Why known compounds were analyzed?

2. What is difference between compound HP7 and HP8?

3. It is difficult to say that IC50 of HP6 and HP7 (14.9 µg/ml and 16.7 µg/ml respectively) are close to IC50 of Cisplatin - positive control (IC50 7.58 µg/ml).

4. From the morphological profile of the cells (Figure 4 and Figure 5) the conclusion for induction of apoptosis cannot be made. More over from the unpublished data of Western blot analysis the authors did not find such kind of activity (line 333).

5. Figure 6 shows the diminishing effects of HP1, HP6 and HP7, but the graphs show that only in the case of compound HP7 such a trend is shown. Please explain other results: the increasing DNA biosynthesis after treatment with HP1 and HP6 compound.

Author Response

A point-by-point response to the Reviewer #2

In the presented MS authors investigated the effect of new polymethoxyflavones isolated from Hottonia palustris on the oral squamous carcinoma (SCC-25) cell line. The study is developed in a current scientific field with great potential for future work. I have some questions pointed to the authors:

  1. The authors analyzed seven compounds four of them are new and three are known. Why known compounds were analyzed?

RESPONSE: The known compounds 1-3 were described for the first time as isolated from the H. palustris herb. The presence of zapotin, found in this species in our previous report was now quantified along with the other compounds in extracts from HP for the first time. In this way, it contributes to the phytochemical characteristics and provides the basis for presuming its potential biological activity, and confirmed chemotaxonomic belonging to the Primulaceae family (primuletin, zapotin).

  1. What is difference between compound HP7 and HP8?

RESPONSE: The indicated chloroform extract (HP7) and chloroform fraction (HP8) showed mainly differences in the quantitative composition of the tested compounds. Figure S34 also showed qualitative differences (Rt 5.5-9 min.). They were included to the manuscript to maintain the qualitative and quantitative control of the complete extraction process. A scheme of obtaining HP8 was retained for complete purification from matrix residual lipophilic compounds for further selective fractionation.

  1. It is difficult to say that IC50 of HP6 and HP7 (14.9 µg/ml and 16.7 µg/ml respectively) are close to IC50 of Cisplatin - positive control (IC50 7.58 µg/ml).

RESPONSE: According to the many tested samples, the influence of HP6 and HP7 on cell line showed effects relatively close to the positive control. However, as suggested Reviewer, the sentence has been detailed and corrected (lines 309-314).

  1. From the morphological profile of the cells (Figure 4 and Figure 5) the conclusion for induction of apoptosis cannot be made. More over from the unpublished data of Western blot analysis the authors did not find such kind of activity (line 333).

RESPONSE: The assumptions contained in the manuscript have been corrected to clarify the results (lines 325, 341-344).

  1. Figure 6 shows the diminishing effects of HP1, HP6 and HP7, but the graphs show that only in the case of compound HP7 such a trend is shown. Please explain other results: the increasing DNA biosynthesis after treatment with HP1 and HP6 compound.

RESPONSE:  The increasing DNA biosynthesis after treatment with HP1 and HP6 extracts were shown in the low concentration of used samples (6.25 µg/mL and 12.5 µg/mL) however higher doses showed satisfactory results in DNA biosynthesis of SCC cells after 24 as well as 48 h of incubation. As the Reviewer noticed only elevated (HP1 > 50 µg/mL and HP6 > 25 µg/mL) concentration of DNA biosynthesis was lowered and following this line it should be taken into consideration in our further studies. Also, we wish to apologize to the Reviewer for confusing statistical significance marks in figures 6 and 7. Actually, the observed increase in DNA biosynthesis for low concentrations of HP1 is not significant. Both figures were corrected.

Reviewer 3 Report

Molecules (Manuscript ID: molecules-1784808), Comments to the Authors:

Title: New polymethoxyflavones from Hottonia palustris evoke DNA biosynthesis-inhibitory activity in an oral squamous carcinoma (SCC-25) cell line

Comments

The submitted manuscript discussed the isolation of four new compounds, 5-hydroxy-2’,6’-dimethoxyflavone (4), 5-hydroxy-2’,3’,6’-trimethoxyflavone (5), 5-dihydroxy-6-methoxyflavone (6), and 5,6’-dihydroxy-2’,3’-dimethoxyflavone (7), and three known compounds, 1,3-diphenylpropane-1,3-dione (1), 5-hydroxyflavone (2), and 5-hydroxy-2’-methoxyflavone (3), from the aerial parts of Hottonia palustris. Their chemical structures were determined using spectroscopic and crystallographic methods. The quantitative analysis of the compounds (1-7) and the zapotin (ZAP) in methanol (HP1), petroleum (HP6), and two chloroform extracts (HP7 and HP8) were also determined using HPLC-PDA. The biological activity of these compounds and extracts on the oral squamous carcinoma cell (SCC-25) line was investigated by considering their cytotoxic effects using the MTT assay.

Despite the presented results, I think the manuscript cannot be published the results are preliminary and mediocre and do not merit publication. The 4 new compounds are very similar to the known compounds and the cytotoxic activity is mediocre for all compounds. I think the authors should try computational methods to find the right targets for their compounds and test other types of cancer cells. 

Author Response

A point-by-point response to the Reviewer #3

  1. Despite the presented results, I think the manuscript cannot be published the results are preliminary and mediocre and do not merit publication. The 4 new compounds are very similar to the known compounds, and the cytotoxic activity is mediocre for all compounds. I think the authors should try computational methods to find the right targets for their compounds and test other types of cancer cells.

RESPONSE: We would like to thank the Reviewer for the extremely critical comment (this is not an evaluation report) regarding the substantive improvement of our manuscript. The proposed computational methods and the search for new goals are valuable suggestions for further research. While respecting the opinion expressed, we cannot agree with the allegations of poor-quality research or its preliminary stage. Our study reports on four completely new compounds found in the plant kingdom. In addition, these compounds were obtained in the process of very labour-intensive isolation and crystallization, and their chemical structure was confirmed by numerous spectral techniques, including UV-Vis, HRMS, 1D and 2D NMR and crystallographic evaluation. In addition, the manuscript describes, for the first time in the world, a new, validated HPLC-PDA analytical method, which allows for the qualitative and quantitative characterization of the tested extracts from H. palustris and species from the Primulaceae family, where primuletin and zapotin can be used as markers. Ultimately, the studies showed the cytostatic activity of the selected compounds, but more importantly, the selectivity of the action on normal fibroblast cells. In conclusion, we believe that our manuscript requires reliable, careful, in-depth reflection, and it does not deserve such a negative assessment.

Round 2

Reviewer 2 Report

I

I accept the replies to my remarks as complete and support the publication of the article.

Author Response

Thank you for your positive opinion on our manuscript.

Reviewer 3 Report

Molecules (Manuscript ID: molecules-1784808), Comments to the Authors:

Title: New polymethoxyflavones from Hottonia palustris evoke DNA biosynthesis-inhibitory activity in an oral squamous carcinoma (SCC-25) cell line

Comments

I read the author’s response to my comments, and I believe they did not respond to the main concern regarding their work. Only 2 of the isolated compounds showed mediocre cytotoxic results against a single cancer cell line. Other compounds were inactive. What are the future perspectives of these compounds as cytotoxic agents? The differences between the new and known compounds are only in the positions of the methoxy groups substitution. I think this work should be further elaborated with more in-depth experiments to merit publication.  

Author Response

While respecting again, the opinion expressed, we cannot agree with the allegations of poor-quality research or its preliminary stage. Our study reports on four completely new compounds found in the plant kingdom. In addition, these compounds were obtained in the process of very labour-intensive isolation and crystallization, and their chemical structure was confirmed by numerous spectral techniques, including UV-Vis, HRMS, 1D and 2D NMR and crystallographic evaluation. In addition, the manuscript describes, for the first time in the world, a new, validated HPLC-PDA analytical method, which allows for the qualitative and quantitative characterization of the tested extracts from H. palustris and species from the Primulaceae family, where primuletin and zapotin can be used as markers. Ultimately, the studies showed the cytostatic activity of the selected compounds, but more importantly, the selectivity of the action on normal fibroblast cells. In conclusion, we believe that our manuscript requires reliable, careful, in-depth reflection, and it does not deserve such a negative assessment.